# Functional Ingredients and Food Preservative in Immature Persimmon "Tekka-Kaki"

Akiyoshi Sawabe [1,2,*], Natsumi Ohnishi [2], Sachiko Yoshioka [1], Kunihiro Kusudo [1], Kenichi Kanno [3] and Yoshiyuki Watanabe [4,5]

[1] Department of Applied Biological Chemistry, Faculty of Agriculture, Kindai University, Nakamachi, Nara 631-8505, Japan; yoshioka@nara.kindai.ac.jp (S.Y.); kunicenter@gmail.com (K.K.)

[2] Graduate School of Agriculture, Kindai University, Nakamachi, Nara 631-8505, Japan; ohnishi.n@tamura-p.co.jp

[3] Department of Biological and Environmental Chemistry, Kindai University, Iizuka, Fukuoka 820-8555, Japan; kanno@fuk.kindai.ac.jp

[4] Department of Biotechnology and Chemistry, Faculty of Engineering, Kindai University, Takayaumenobe, Higashihiroshima 739-2116, Japan; watanabe@biochem.osakafu-u.ac.jp

[5] Graduate School of Life and Environmental Sciences, Osaka Prefecture University, 1-1 Gakuen-cho, Nakaku, Sakai-shi, Osaka 599-8531, Japan

* Correspondence: sawabe@nara.kindai.ac.jp

**Abstract:** Immature persimmons are unripe fruits that are cut off during the persimmon cultivation process and immediately discarded, amounting to an annual fruit loss of approximately 100 to 400 kg per 1000 m$^2$. The purpose of this study was to make effective use of unused resources, namely, immature persimmons, and attempt to use them as food additives. In this study, we studied the Tone Wase (fully astringent persimmon) and Fuyu (fully sweet persimmon) cultivars. As a result, we performed a component analysis of the immature persimmons, isolating 12 compounds, of which two were newly identified. Differences in the components and their contents were found between cultivars and between the peel and flesh. To effectively use immature persimmons as food for the elderly, we searched for active substances that inhibit AGE formation and found that extracts of immature persimmons and isolated compounds showed high activity. In particular, high activity was observed for catechin and its polymeric form, procyanidin. Regarding the inhibition of aroma deterioration, 5 mg/L of gallic acid in octadecane was found to be the optimal condition for the inhibition of citral deterioration. As for antimicrobial activity, we found that extracts at a concentration of 500 mg/L had no antimicrobial effect. Based on these findings, we made a microencapsulation process, and plan to advance to the clinical trial study in future. These findings confirmed the effectiveness of immature persimmons, which are an unused resource, and reveal their potential as a food for the elderly and as a food additive in other food products, which we hope will lead to new industrial innovations.

**Keywords:** functional ingredients; food preservative; immature persimmon; citral; AGE inhibitory activity; deterioration inhibitory effect

## 1. Introduction

The scientific name of the oriental persimmon (hereinafter referred to as "persimmon") is *Diospyros kaki*, and it is classified as a deciduous tree belonging to the family Ebenaceae of the genus *Diospyros*. The persimmon originates from northern China and is said to have been introduced to Japan during the Nara period, which took place over the years 710–794. The astringency of the peels of persimmon is caused by tannin, which is a condensation product of (-)-epigallocatechin, (-)-epicatechin, and their galloyl compounds. When persimmons ripen, acetaldehyde is generated within the flesh of the fruit and forms cross-links with the water-soluble tannin to form polymers, making the tannin insoluble and thus reducing the persimmon's astringency [1]. In addition, it is said that

when persimmons ripen and become sweet, the abovementioned water-soluble tannin is further oxidatively polymerized by the action of polyphenol oxidase and becomes insoluble polymers, which reduce the astringency of the fruit.

There are more than 800 to 1000 cultivars of persimmon in Japan, but only 10 well-known cultivars, such as the 'Fuyu' and 'Hira Tanenashi', which are widely cultivated throughout the country. Among these cultivars, Tone Wase and Fuyu were used in this study.

At harvest time, Tone Wase, which is a partially astringent persimmon, contains 1% to 2% soluble tannins and is intensely astringent. Anaerobic treatment with carbon dioxide or ethanol increases the acetaldehyde content, allowing the removal of its astringency (deastringency).

Meanwhile, Fuyu (also known as Mizugosho) was first reported by Dr. Onda in 1902, and since then has rapidly spread. The Fuyu's tree is robust and produces large fruits; however, excessive fruiting causes the lateral branches to droop, which in turn results in poor fruit growth and an increase in the proportion of small fruit. Consequently, it is necessary to regulate fruiting, mainly through disbudding. The ripening period varies by region, but in the Kinki region of Japan, it is from late October to mid-November.

The process of growing persimmons is shown in Figure 1, and fruit thinning, which is the process of cutting off unripe fruit, is done in July and August. The unripe fruits that are cut off are called immature persimmons, or 'Tekka-kaki'. The immature persimmons are left on the ground after being cut off and can be considered a discarded, unused resource. Therefore, the purpose of this study was to investigate "the effective use of immature persimmons".

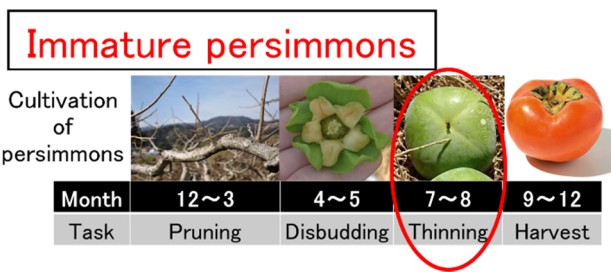

**Figure 1.** Cultivation of persimmons.

An example of the use of immature fruit includes unripe *Citrus unshiu*. The extract of unripe *Citrus unshiu* has shown anti-allergic effects in in vivo studies using type I allergy models (PCA reaction test) and type IV allergy models (PC-CD). The relationships between fruit ripeness and the intensity of the anti-type I and anti-type IV allergic effects were investigated, and it was reported that these effects were stronger in unripe fruit and weakened as the fruit ripened. This effect has gained attention, resulting in the use of unripe *Citrus unshiu* as a dietary supplement [2,3].

One of the best examples of the use of persimmons in processed foods includes the use of dried persimmons; however, compared to other fruits, their application is limited. Furthermore, the processing of immature persimmons into dried products has the potential of minimizing the associated labor cost, as it does not directly generate income and the fruit that have already grown to a certain size are cut off before harvesting. If immature persimmons could be used for processed foods, it is expected that it would increase the motivation of persimmon growers because it would be possible to generate income from that work, and it would also increase the consumption of processed persimmons.

In this study, we used immature persimmons as food for the elderly. Currently, the proportion of the elderly among the Japanese population is as high as one in three, and the food demand for the elderly is increasing. The elderly tend to show signs of difficulty with ingestion and swallowing as they age [4]. As a result, this can lead to problems such as "loss of the pleasure of eating", "decreased function of the digestive system", "malnutrition such as energy malnutrition", and "induced aspiration pneumonia". Therefore, we are

developing high-value-added microcapsules with the goal of providing processed foods with moderate elasticity that are nutritionally balanced and flavored.

The first step in the microencapsulation process is to emulsify a two-phase solution consisting of an oil phase and a water phase containing a gelling agent. At this stage, flavoring is added to the oil phase as a secondary function, and functional components are added to the water phase as a tertiary function. These are then used to make emulsions, which are then dried and microencapsulated. Regard for functional components is necessary, as they are intended for the elderly, prevention of aging, and age-related diseases such as skin aging, arteriosclerosis, osteoporosis, and diabetic complications. Therefore, we searched for AGE formation inhibitors in immature persimmons. The term "AGEs" is a general term for a group of structures produced by glycation reactions and does not indicate a fixed structure. Various structures of AGEs have been elucidated to date, including those produced in vitro. AGEs are characterized by their yellowish-brown color, fluorescence, and the formation of cross-links between proteins. In recent years, AGEs have been suggested to be involved in diabetic complications, atherosclerosis, tumor growth, metastasis, inflammatory reactions, osteoporosis, and skin aging [5]. Therefore, we focused on anti-glycation and conducted tests to inhibit AGE formation to prevent the progression of disease and aging.

Regarding flavoring and fragrances, citral, a citrus fragrance, was used to mask odors and add flavoring because when the food is ground into a gel, its flavor degrades. The flavoring and fragrance industry has been expanding annually, and the global market for food flavoring agents is estimated to be 1.5 trillion yen [6]. Flavoring and fragrances generally contribute to the taste and flavor of foods. The citral used in this study is also widely used as a flavoring agent in citrus fruit juice beverages and foods [7].

However, odor problems (off-flavors) may occur during the process from food production to actual consumption. Off-flavors refer to the "odor" perceived by the addition of odorous components not originally present in the food, the increase or decrease of some odorous components present in the food, or the change in the overall balance of the odorous components [8]. The elucidation of the conditions for the formation of off-flavors and the study of the methods to control them may lead to improvement in food processing technology and ultimately to improvement in food quality.

Citral is a $C_{10}H_{16}O$ monoterpene composed of two isoprenes. Its structure has a cis-trans structural isomer, which refers to a compound that has two isomers: geranial (E-isomer) and neral (Z-isomer). Citral is an essential oil component abundant in lemongrass and lemon balm and is used as a component of citrus fragrances because of its strong lemon-like scent. It is used as a flavoring agent in cosmetics and foods and as a raw material in the manufacture of other flavors and fragrances. The bioactivity of citral is reported to have antimicrobial, anti-inflammatory, analgesic, and sedative effects [9].

Citral is easily oxidized under acidic and oxidative stress conditions, resulting in not only the loss of its lemon scent but also the production of undesirable off-flavors. In addition, citral gradually changes from a pale yellow to stronger yellowish color as it deteriorates. The mechanism of the formation of off-flavors in citral, as shown in Figure 2, is described as follows: First, p-menthadien-8-ol is formed by cyclization, which is then oxidized to p-cymen-8-ol, followed by the formation of p-methyl acetophenone via $\alpha$-p-dimethyl styrene. This p-methyl acetophenone has a lower threshold than the other products (2.7–10.8 ng/L in air), a very strong deterioration odor, and is known to cause a significant quality loss in citrus products [10]. The compound, p-menthadien-8-ol, has a lemon aroma, but compounds of the latter structures cause the deterioration of odors similar to p-methyl acetophenone.

**Figure 2.** Formation route of deteriorated substances of citral.

The addition of various antioxidants to inhibit the formation of compounds associated with citral-derived deterioration odors was investigated. However, a method that can sufficiently inhibit the formation of deteriorated substances has not yet been developed. As a result, we attempted to evaluate immature persimmons, which are an unused resource that has yet to be exploited.

## 2. Materials and Methods

### 2.1. Materials

We used the immature persimmon which we collected in Nara, Japan: *Diospyros Kaki* 'Tone Wase' (6.50 kg) from an astringent persimmon and *Diospyros Kaki* 'Fuyu' (6.33 kg) from a sweet persimmon. It was divided into peels and fruity flesh. The abbreviated designations are as follows: Tone Wase-peel (TP); Tone Wase- fruity flesh (TF); Fuyu-peel (FP); and Fuyu- fruity flesh (FF). We also used the ripe persimmon for a comparison experiment. Citral, dodecane, tetradecane, octadecane, and all the other chemicals were purchased from Wako Pure Chemical Industries, Ltd. (Osaka, Japan).

Decaglycerol monolaurate SY-Glyster ML-750 (ML-750) was supplied by Sakamoto Yakuhin Kogyo Co., Ltd. (Osaka, Japan).

### 2.2. Isolation of Functional Compounds

Used persimmons were obtained by separating the peel and flesh of each. Their immature persimmon was chopped by a commercial blender and homogenized after, finishing with a total volume of 3 L with water. Cold methanol (7 L) was added to the water solution, and the mixture was allowed to stand 1 week in the dark to obtain crude extracts. The crude extracts were obtained by filtration and vacuum concentration. The crude extracts were extracted with hexane, then with 1-butanol to obtain hexane, 1-butanol, and water extracts. The isolation of 1-butanol extracts was performed by repeated silica gel chromatography and gel filtration to obtain a total of 12 isolated compounds.

### 2.3. Gel Filtration

Gel filtration was carried out in prepacked column (2.5 cm × 30 cm) of HW-65F TSK gel (Tosoh Co., Ltd., Tokyo, Japan) under medium pressure (flow rate, 1 mL/min), and absorptions of the eluates were measured at 280 nm with an UVICON UV-2800 instrument (Advantec Toyo Co., Ltd., Tokyo, Japan). The column for the extract was eluted with distilled water and then successively with 50% methanol. The eluate was collected in every min, and each fraction was concentrated in vacuo and freeze-dried.

### 2.4. Column Chromatography on Silica Gel

Each fraction obtained by gel filtration was chromatographed over silica gel (C-300 Wako gel, Wako Pure Chemical Industries Ltd., Osaka, Japan; open column, 2.5 cm × 30 cm) with

chloroform-methanol-water (60:29:3 *v*/*v*/*v*) or chloroform-methanol (5:1 or 8:1 *v*/*v*) as the eluent.

### 2.5. Mass Spectrometry and NMR Analyses

　　MS spectra were obtained with JEOL JMX-HX 100 and JMA-DA 5000 spectrometers under Xe bombardment (6.0 keV) and an LCMS-2020 mass spectrometer (Shimadzu instrument, Kyoto, Japan). $^1$H-NMR (500 MHz or 270 MHz), $^{13}$C-NMR (125.7 MHz or 67.5 MHz) and 2D NMR spectra were obtained with BRKER AVANCE$^{TM}$ III NanoBay (500 MHz or 400 MHz) spectrometers in a 5 mmφ tube at various temperatures. A 15 mg of **5** was used for 2D NMR (HMBC). Chemical shifts were reported as parts per million (ppm) downfield from internal tetramethylsilane as the standard. An HPLC (GL-7400 with PDA Detector; GL Science Inc., Tokyo, Japan) equipped with Cosmosil packed column Cholester (5 um, 25 cm × 4.6 mm I.D.) was used. Analysis conditions were as follows: eluate, 3% acetic acid/MeOH (70/30: *v*/*v*); flow rate, 1.0 mL/min; column temperature, 40 °C; detection wavelength, UV 270 nm; injection volume, 10 μL.

### 2.6. Isolated Compounds

**4-(2-*O*-β-glucopyranosyl ethyl)phenol (1)**
FAB-MS: *m/z* 299 [M-H]$^-$
$^1$H-NMR (CD$_3$OD, δppm): 2.85(2H, m), 3.41(1H, m), 3.50(2H, m), 3.61(1H, m), 3.81(1H, m), 4.83(2H, d, J = 9.5 Hz), 6.66(2H, d, J = 8.5 Hz), 7.03(2H, d, J = 8.5 Hz). 76.7
$^{13}$C-NMR (CD$_3$OD, δppm): 30.99, 61.81, 71.04, 74.11,2, 79.10, 82.69, 104.37, 116.10(×2), 130.39(×2), 133.82, 156.46.

**Gallic acid (2)**
FAB-MS: *m/z* 169 [M-H]$^-$
$^1$H-NMR (CD$_3$OD, δppm): 7.03(2H, s).
$^{13}$C-NMR (CD$_3$OD, δppm): 110.14, 126.83, 137.94, 146.03, 167.69.

**3β-hydroxy-olean-12,18-dien-28-oic acid (3)**
FAB-MS: *m/z* 453 [M-H]$^-$
$^1$H-NMR (CD$_3$OD, δppm): 0.76(3H, s), 0.83(3H, s), 0.93(3H, s), 0.95(3H, s), 0.96(6H, s), 1.11(3H, s), 1.30~1.66(8H, m), 1.91(1H, m), 3.14(1H, m), 5.22(2H, m).
$^{13}$C-NMR (CD$_3$OD, δppm): 16.03, 16.38, 17.65, 17.81, 19.48, 21.57, 24.10, 24.37, 25.32, 26.40, 27.90, 28.78, 29.22, 31.62, 31.77, 33.57, 33.83, 34.03, 34.34, 43.25, 47.63, 54.37, 56.75, 79.70, 123.65, 126.91, 139.64, 145.20, 181.63.

***cis*-ferulic acid (4)**
FAB-MS: *m/z* 193 [M-H]$^-$
$^1$H-NMR (CD$_3$OD, δppm): 3.90(3H, s), 6.19(1H, d, J = 10 Hz), 6.66(1H, d, J = 8 Hz), 6.68(1H, d, J = 2 Hz), 6.75(1H, dd, J = 2, 8 Hz), 2.86(1H, d, J = 10 Hz).

**2,3-dihydro-2-(4′-hydroxy-3′-methoxyphenyl)-3-hydroxymethyl-7-methoxy-5-benzofuran propanol (5)**
FAB-MS: *m/z* 359 [M-H]$^-$
$^1$H-NMR (CD$_3$OD, δppm): 1.82(2H, m), 2.63(2H, m), 3.56(2H, m), 3.75(1H, m), 3.81(3H, s), 3.82(2H, m), 3.85(3H, s), 5.48(1H, d, J = 6 Hz), 6.72(2H, br. s), 6.75(1H, d, J = 8 Hz), 6.82(1H, dd, J = 2, 8 Hz), 6.94(1H, d, J = 2 Hz).
$^{13}$C-NMR (CD$_3$OD, δppm): 32.91, 35.84, 55.48, 56.32, 56.70, 62.22, 64.98, 88.98, 110.47, 114.02, 116.10, 117.90, 119.68, 129.85, 134.80, 136.91, 145.21, 147.48, 147.51, 149.08.

**catechin (6)**
ESI-MS: *m/z* 289 [M-H]$^-$
$^1$H-NMR (CD$_3$OD, δppm): 2.50(1H, dd, J = 8, 16 Hz), 2.84(1H, dd, J = 5.5, 16 Hz), 3.97(1H, m), 4.56(1H, d, J = 7.5 Hz), 5.85(1H, d, J = 2 Hz), 5.92(1H, d, J = 2 Hz), 6.71(1H, dd, J = 2, 8 Hz), 6.75(1H, d, J = 8 Hz), 6.83(1H, d, J = 2 Hz).

**procyanidin B (7)**

ESI-MS: *m/z* 578 [M]$^-$

$^1$H-NMR (CD$_3$OD, δppm): 2.49(1H, dd, J = 8, 16 Hz), 2.84(1H, dd, J = 5.5, 16 Hz), 3.88(1H, m), 3.96(2H, m), 4.55(2H, d, J = 7 Hz), 5.84(1H, d, J = 2 Hz), 5.91(1H, s), 5.92(1H, d, J = 2 Hz), 6.71(2H, dd, J = 2, 8 Hz), 6.75(2H, d, J = 8 Hz), 6.83(2H, d, J = 2 Hz).

**3,3′-*O*-diprotocatechuoyl procyanidin B (8)**

FAB-MS: *m/z* 549 [M-H]$^-$

$^1$H-NMR (CD$_3$OD, δppm): 2.05(1H, m), 2.34(1H, m), 3.81(1H, m), 3.84(2H, m), 5.14(1H, dd, J = 7.5, 15 Hz), 5.24(1H, d, J = 7.5 Hz), 6.20(1H, d, J = 1.5 Hz), 6.39(1H, d, J = 1.5 Hz), 6.40(1H, s), 6.85~6.90(6H, m), 7.59(2H, dd, J = 2, 8.5 Hz), 7.70(1H, d, J = 2 Hz), 7.83(1H, d, J = 2 Hz), 8.05(1H, d, J = 8.5 Hz), 8.08(1H, d, J = 8.5 Hz).

**3,3′-*O*-digalloyl procyanidin B (9)**

FAB-MS: *m/z* 881 [M-H]$^-$

$^1$H-NMR (CD$_3$OD, δppm): 2.49(1H, dd, J = 8, 16 Hz), 2.80(1H, dd, J = 5.5, 16 Hz), 3.96(1H, m), 4.52(1H, d, J = 7 Hz), 4.55(1H, d, J = 7.5 Hz), 5.25(1H, d, J = 7.5 Hz), 5.84(1H, d, J = 2 Hz), 5.91(1H, s), 6.20(1H, d, J = 2 Hz), 6.39(4H, br. s) 6.71(1H, dd, J = 2, 8 Hz), 6.75(1H, d, J = 8 Hz), 6.83(1H, d, J = 2 Hz), 6.86(1H, d, J = 8 Hz), 6.87(1H, d, J = 2 Hz), 6.88(1H, dd, J = 2, 8 Hz).

**3,3′,3″-*O*-tri-3′-methoxy protocatechuoyl procyanidin C (10)**

FAB-MS: *m/z* 1315 [M-H]$^-$

$^1$H-NMR (CD$_3$OD, δppm): 2.70(1H, dd, J = 8, 15 Hz), 2.82(1H, dd, J = 4.5, 15 Hz), 3.81(9H, s), 4.21(1H, m), 4.26(1H, m), 5.78(1H, d, J = 2 Hz), 5.81(1H, d, J = 2 Hz), 6.19(1H, m), 6.20(1H, s), 6.22(1H, s), 6.23(1H, d, J = 7.5 Hz), 6.29(2H, m), 6.71(2H, d, J = 8 Hz), 6.77(4H, m), 6.81(2H, d, J = 8 Hz), 7.03(2H, br. s), 7.21(1H, d, J = 8 Hz), 7.37~7.55(7H, m).

**ursolic acid (11)**

FAB-MS: *m/z* 455 [M-H]$^-$

$^1$H-NMR (CD$_3$OD, δppm): 0.65(3H, s), 0.77(3H, s), 0.79(3H, s), 0.80(3H, s), 0.81(3H, s), 0.84(3H, d, J = 6.5 Hz), 0.89(3H, d, J = 6.5 Hz), 1.26(4H, m), 1.47(2H, m), 1.60(2H, m), 1.82(2H, m), 1.97(2H, m), 2.28(3H, m), 3.51(1H, m), 5.32(1H, d, J = 5 Hz).

$^{13}$C-NMR (CD$_3$OD, δppm): 11.82, 11.98, 18.75, 19.00, 19.36, 19.78, 21.05, 23.04, 24.27, 26.05, 28.21, 29.13, 29.66, 31.52, 31.88, 33.91, 36.11, 36.47, 37.21, 39.74, 42.29, 45.80, 50.10, 56.03, 56.73, 71.80, 121.70, 140.68, 178.70.

**3β-hydroxy-24-hydroxymethyl-ursan-5(6),12-dien-28-oic acid (12)**

FAB-MS: *m/z* 453 [M-H]$^-$

$^1$H-NMR (CD$_3$OD, δppm): 0.74(3H, d, J = 6 Hz), 0.75(3H, s), 0.88(3H, d, J = 6.5 Hz), 0.91(3H, s), 0.96(3H, s), 1.06(3H, s), 1.11(3H, s), 1.30(5H, m), 1.57(6H, m), 1.89(2H, m), 2.29(2H, m), 2.79(1H, m), 3.20(3H, m), 5.23(1H, m), 5.27(1H, m).

$^{13}$C-NMR (CD$_3$OD, δppm): 15.32, 15.56, 17.05, 18.29, 21.16, 23.56, 25.90, 27.20, 27.68, 28.13, 30.67, 32.42, 33.05, 37.07, 38.76, 39.28, 41.11, 41.66, 41.99, 45.89, 46.46, 47.63, 55.22, 77.21, 49.05, 122.67, 125.88, 137.94, 143.56, 178.38.

*2.7. In Vitro Inhibition Test of AGE Generation*

For the mixture of the sample (20 μL), which was adjusted for each concentration, 0.1 mol/L phosphate buffer solution (PBS) (pH 7.4) (500 mL), distilled water (180 μL), 40 mg/mL of Bovine serum albumin (BSA, Sigma Chemical Co., Ltd., MO, USA) (200 mL), and 2 mmol/L of glucose aqueous solution (100 μL) were stirred. We prepared 2 samples of the same concentrations to see the difference in incubation. Furthermore, as a blank (controlled trial), we used methanol instead of a sample. Each sample was incubated for 30 h at 60 °C (A) and subsequently, 25 °C (B). After incubation, trichloroacetic acid (100 μL) was added to the mixture and stirred into it. Then the mixtures were centrifuged at 4 °C, under the condition of 15,000 rpm for 4 min. The precipitates (AGEs) were dissolved with 1 mL of 0.25 N sodium hydroxide water solution-PBS and 200 μL was poured into a white

microplate. The AGE-derived fluorescence was measured using microplate reader TECAN F200 (Tecan Group Ltd., Zurich, Swiss Confederation) at an excitation wavelength if 360 nm and a fluorescent wavelength of 440 nm. Percentage inhibition of AGEs generation was calculated as,

AGE inhibition rate (%) = {(blank A − blank B)-(sample A − sample B)/(blank A − blank B)} × 100

### 2.8. Preparation of Emulsion

The oil mixture was prepared by mixing 0.05 mL of citral and 0.45 mL of alkane. The mixture was emulsified using 4.5 mL of 1% (*w/v*) SY-Glyster ML-750 as an emulsifier solution [11]. The prepared emulsion was stored at 37 °C and was periodically sampled. For the freezing and thawing experiments, the emulsion was frozen for 3 days at −20 °C and further thawed for 10 min at 30 °C.

### 2.9. Measurement of the Amount of Remaining Citral and Other Compounds in the Emulsion

The emulsion, sampled before and after storage, was extracted with 1 mL of dichloromethane 3 times. The amount of remaining citral and other compounds in 1 μL of the sample was measured by gas chromatography (GC-14B, Shimadzu Corporation, Kyoto, Japan) containing a DB-1 capillary column (30 m × 0.32 mm I.D., 1.00 μm, Agilent J&W, Santa Clara, CA, USA) and a flame ionization detector. The temperatures of the injector and the detector were 220 °C and 240 °C, respectively. The column temperature was maintained at 50 °C for 2 min and raised 5 °C/min and maintained at 180 °C for 2 min.

### 2.10. Minimum Inhibitory Concentration (MIC) Assay

MIC values were determined by the broth micro-dilution method. A 96-well plate was used for this assay, whereby each of well plate was loaded with the inoculums grown to an exponential phase containing $10^7$ CFU/mL of bacteria. An amount of 500 μg/mL of sample was the highest concentration used in this study. Two-fold serial dilution was performed by transferring 100 μL from the highest concentration of treated culture into the next well so that the final volume of each well was 200 μL. Finally, all of the tested plates were incubated.

## 3. Results

### 3.1. Analysis and Comparison of Components in Immature Persimmons

The persimmons used in this study were from two cultivars, 'Tone Wase' and 'Fuyu', and four raw materials were obtained by separating the peel and flesh of each. Each of the four raw materials was macerated in 70% methanol for 1 week to obtain crude extracts. This was followed by the sequential distribution of hexane and 1-butanol to obtain hexane, 1-butanol, and water extracts. The amounts of extracts from immature and ripe persimmons are shown in Table 1. However, hexane extraction from the flesh of immature Tone Wase persimmons (TF) proved to be difficult, and therefore only 1-butanol extraction was performed. Only 1-butanol and water extracts were obtained.

**Table 1.** Amounts of immature persimmon samples and extracts.

| Sample | Quantity | Crude Extract (A) | Hexane Extract (H) | Butanol Extract (B) | Water Extract (W) |
|---|---|---|---|---|---|
| Tone Wase peel (TP) | 1.61 kg | 131.19 g | 0.38 g | 19.89 g | 83.51 g |
| Tone Wase flesh (TF) | 3.40 kg | 403.70 g | − [1] | 12.38 g | 348.96 g |
| Fuyu peel (FP) | 1.06 kg | 77.03 g | 0.08 g | 3.95 g | 101.60 g |
| Fuyu flesh (FF) | 4.73 kg | 335.88 g | 0.06 g | 16.45 g | 331.76 g |

[1] −: Butanol extraction without extraction.

The isolation of 1-butanol extracts was performed by repeated silica gel chromatography and gel filtration. As a result, we succeeded in isolating seven compounds, **1, 2, 3, 4, 5, 8**, and **10**, from the Tane Wase cultivar, and five compounds, **6, 7, 9, 11**, and **12**, from the Fuyu cultivar, making a total of 12 isolated compounds (Figure 3).

**Figure 3.** Isolated compounds.

The MS and NMR spectral data for compounds **1**, **2**, and **4–11** agreed with those of the authentic 4-(2-*O*-β-glucopyranosyl ethyl)phenol, gallic acid, *cis*-ferulic acid, 2,3-dihydro-2-(4′-hydroxy-3′-methoxyphenyl)-3-hydroxymethyl-7-methoxy-5-benzofuran propanol, catechin, procyanidin B, 3,3′-*O*-diprotocatechuoyl procyanidin B, 3,3′-*O*-digalloyl procyanidin B, 3,3′,3″-*O*-tri-3′-methoxy protocatechuoyl procyanidin C and ursolic acid.

Compound **3** was isolated from the Tane Wase cultivar. The molecular formula of **3** was found to be $C_{30}H_{46}O_3$ by FAB-MS, which showed a characteristic peak at *m/z* 453 [M − H]$^-$. The $^1$H-NMR and $^{13}$C-NMR spectral data of **3** closely resembled those of oleanolic acid. However, the location of the olefin proton was verified by HMBC, thus confirming the location of olefin to a C-18–C-19. Based on this evidence, the structure of compound **3** was determined to be 3β-hydroxy-olean-12,18-dien-28-oic acid.

Compound **12** was isolated from the Fuyu cultivar. The molecular formula of **12** was found to be $C_{30}H_{46}O_3$ by FAB-MS, which showed a characteristic peak at *m/z* 453 [M − H]$^-$. The $^1$H-NMR and $^{13}$C-NMR spectral data of **12** closely resembled those of compound **11**. The locations of the hydroxymethyl group and olefin were identified to be C-4 and C5–C6, respectively, by comparing those data. Therefore, the structure of compound **12** was proved to be 3β-hydroxy-24-hydroxymethyl-ursan-5(6),12-dien-28-oic acid.

Among them, compounds **3** and **12** are new compounds that have not yet been published in the literature. Moreover, the distributions and yields of the crude extracts of different cultivars in the peel and flesh of immature persimmons, using HPLC and LC-MS, are shown in Table 2. In summary, differences were found in the components and their contents between cultivars and between the peel and flesh.

**Table 2.** Distribution and yields of isolated compounds (crude extracts).

| Compound | Tone Wase Peel (TP) | Tone Wase Flesh (TF) | Fuyu Peel (FP) | Fuyu Flesh (FF) |
|---|---|---|---|---|
| 1 | 1.66 | 4.85 | 1.77 | 1.67 |
| 2 | 1.11 | 0.71 | 0.19 | 0.09 |
| 3 | 2.74 | – [1] | 0.89 | – [1] |
| 4 | 0.13 | 0.49 | 0.06 | 0.06 |
| 5 | 0.35 | – [1] | 1.53 | – [1] |
| 6 | 1.49 | 0.95 | 3.83 | 1.75 |
| 7 | 29.69 | 40.68 | 2.57 | 1.37 |
| 8 | 0.59 | 2.20 | – [1] | – [1] |
| 9 | – [1] | 0.96 | 1.25 | 0.87 |
| 10 | – [1] | 0.42 | – [1] | – [1] |
| 11 | – [1] | – [1] | – [1] | 0.01 |
| 12 | – [1] | – [1] | 0.01 | – [1] |

[1] –: Not detected. Unit: %.

### 3.2. AGE Inhibitory Activity and Cytotoxicity Test

The IC50 of the AGE inhibitory activity test for each extract of immature and ripe persimmons is shown in Table 3. The $IC_{50}$ of aminoguanidine, a positive control, was 31.1 mg/L, whereas the crude extracts of immature persimmons showed equal or higher activity.

**Table 3.** $IC_{50}$ values of the AGE inhibitory activity test of each extract.

| Sample | Tone Wase Peel (TP) | Tone Wase Flesh (TF) | Fuyu Peel (FP) | Fuyu Flesh (FF) |
|---|---|---|---|---|
| Crude extract (A) | 35.0 | 32.2 | 6.5 | 20.8 |
| Hexane extract (H) | 15.5 | – [1] | 455.3 | 95.5 |
| Butanol extract (B) | 14.4 | 12.1 | 17.0 | 42.5 |
| Water extract (W) | 52.7 | 31.9 | 23.1 | 28.8 |
| Ripe persimmons (m-) | 18.6 | 5.8 | 38.6 | 489.8 |
| Ripe persimmons (astringency removed) (m-CO$_2$) | 147.6 | 120.6 | – [1] | – [1] |
| Aminoguanidine (positive control) | | 31.1 | | |

[1] –: No sample. Unit: mg/L.

Among the other extracts of immature persimmons, 1-butanol extracts were found to be the most active. In comparison with ripe persimmons, the activity in Tone Wase persimmons that had not undergone astringency removal was higher than that in immature persimmons. However, this activity was greatly reduced by astringency removal. Regarding Fuyu persimmons, activity in ripe persimmons was lower than that in immature persimmons.

The results of the AGE formation inhibitor activity tests of compounds **1** to **12** are shown in Table 4. All the compounds showed higher activity than aminoguanidine. In particular, phenolic compounds **6** to **10** showed very high activity. These compounds were found to contribute to the remarkable activity of the extracts. In addition, no toxicity was observed in the cytotoxicity test (data are not shown), and there were no problems with their application to food. In summary, high activity was exhibited by both extracts and isolates, and high activity was observed for phenolic compounds with a catechin structure, such as catechin and procyanidin.

**Table 4.** IC$_{50}$ values of the AGE inhibitory activity test of the isolated compounds.

| Compound | AGE Inhibitory Activity IC$_{50}$ Values |
|---|---|
| Aminoguanidine (positive control) | 420.0 |
| 1 | 313.9 |
| 2 | 35.7 |
| 3 | 174.2 |
| 4 | 410.0 |
| 5 | 395.8 |
| 6 | 4.7 |
| 7 | 4.5 |
| 8 | 10.2 |
| 9 | 7.2 |
| 10 | 4.5 |
| 11 | 330.1 |
| 12 | 1026.6 |

Unit: μM.

### 3.3. Beverages and Flavoring

Emulsions, which are thermodynamically metastable dispersion systems, are one of the forms of application of flavors in foods. Emulsions can change the rate of chemical reactions due to the presence of interfacial films that inhibit the reaction between the lipid and water phases [12]. The deterioration of citral may occur under both acidic and oxidative conditions, suggesting that the stability of citral may be improved by the formation of an emulsion. In this study, three types of alkanes (dodecane, tetradecane, and octadecane), with citral mixed into them, were used for the oil phase, while SY-Glyster ML-750, an emulsifier, was used for the water phase. SY-Glyster ML-750 is a polyglyceryl fatty acid ester made by esterifying polyglycerin-10 and lauric acid. SY has been used as an excellent emulsifier in whipped cream, coffee whiteners, margarine, shortening, chocolate, milk drinks, and functional lipids. In this study, we investigated the optimal alkane and the inhibitory effect of citral deterioration caused by thinned persimmons in an emulsified form.

Citral is a combination of the isomers geranial and neral. It is a lemon-scented aroma component that is widely used as a fragrance and flavoring in cosmetics and food. However, citral is susceptible to oxidation and deterioration, which causes the loss of its lemon scent, leading to the deterioration of food quality. Therefore, the addition of antioxidants is necessary.

In this study, myrcenol and linalooloxide were detected as deteriorated substances using GC-MS, but other deteriorated substances were not tested. This formation route is thought to be caused by the effect of the SY-Glyster emulsifier (Figure 4). Subsequently, we quantified citral, myrcenol, and linalooloxide.

We used an alkane–citral mixture as the oil phase and three types of alkanes (dodecane, tetradecane, and octadecane). Then, in the water phase, the SY-Glyster ML-750 solution was used as an emulsifier, and TPA, a crude extract of the peel of Tone Wase persimmons, was added as an antioxidant, and gallic acid was added as a comparator substance at the respective concentrations. The control was prepared without the antioxidant TPA or GA. All of the abovementioned components were emulsified to produce emulsions, which were stored at 37 °C for 4 weeks, and the amounts of citral and deteriorated substances were measured and compared by gas chromatography every week. In this experiment, we investigated the oxidative stability of citral induced by the addition of antioxidants and the optimal combinations between the different types of alkanes and antioxidant concentrations.

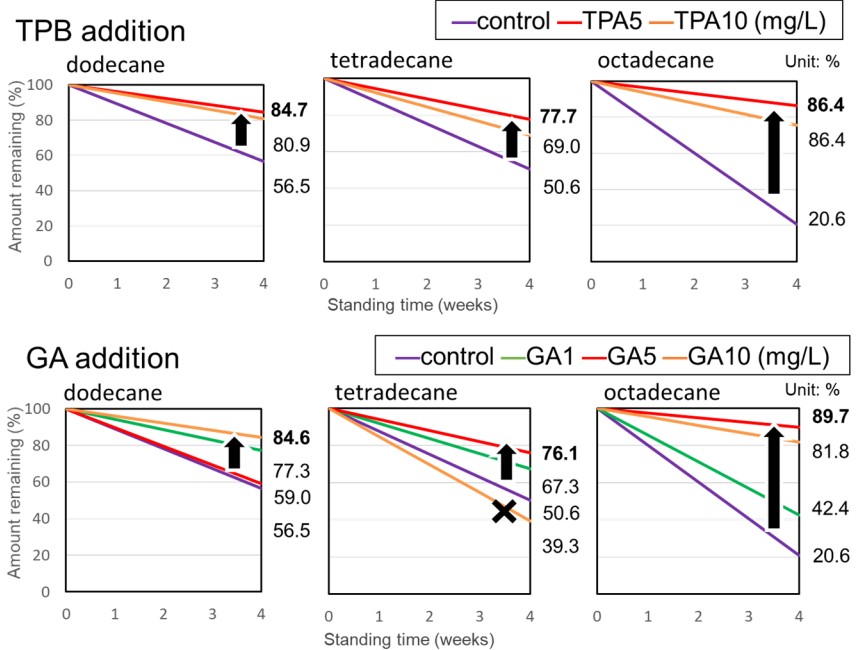

**Figure 4.** Formation route of deteriorated substances myrcenol and linalool oxide.

Figure 5 shows the graphs depicting the change in the amount of remaining citral. The upper row shows the extract (TPA), and the lower row shows gallic acid (GA). The vertical axis shows the amount remaining (unit: %), and the horizontal axis represents the standing time (unit: weeks). In the graph, the control is shown in purple, and antioxidant concentrations are shown in green for 1 mg/L, red for 5 mg/L, and orange for 10 mg/L. The number next to the graph shows the amount remaining after 4 weeks. The results demonstrate that almost all conditions exceed the amount remaining in the control, inhibiting the decrease in citral. The optimal condition was GA 5 mg/L in octadecane, which had the highest amount remaining.

**Figure 5.** Changes in the amount of citral.

Figure 6 shows the graphs of myrcenol, a deteriorated substance. This graph follows a projection similar to the previous one. The results show that almost all conditions are below the formation rate of the control, suppressing the increase in myrcenol. The optimal condition was GA 5 mg/L octadecane, which had the lowest formation rate.

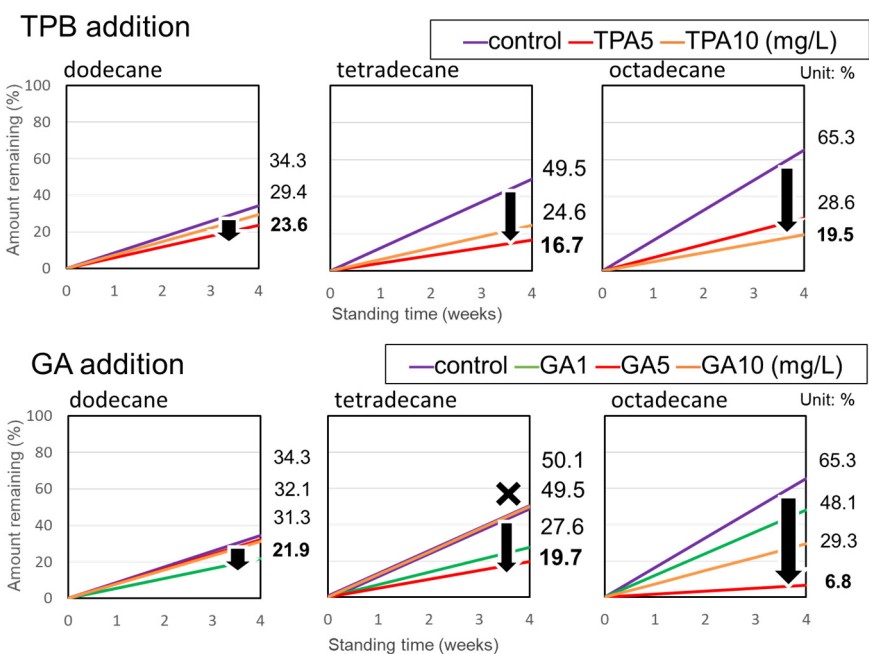

**Figure 6.** Changes in the amount of myrcenol.

Figure 7 shows graphs of linalooloxide, which is another deteriorated substance. The results equally show that almost all conditions are below the formation rate of the control, inhibiting the increase in linalooloxide. The optimal condition was GA 5 mg/L in octadecane, which had the lowest formation rate. In summary, the optimal condition that best inhibited both the decrease in citral and the increase in deteriorated substances was GA 5 mg/L in octadecane.

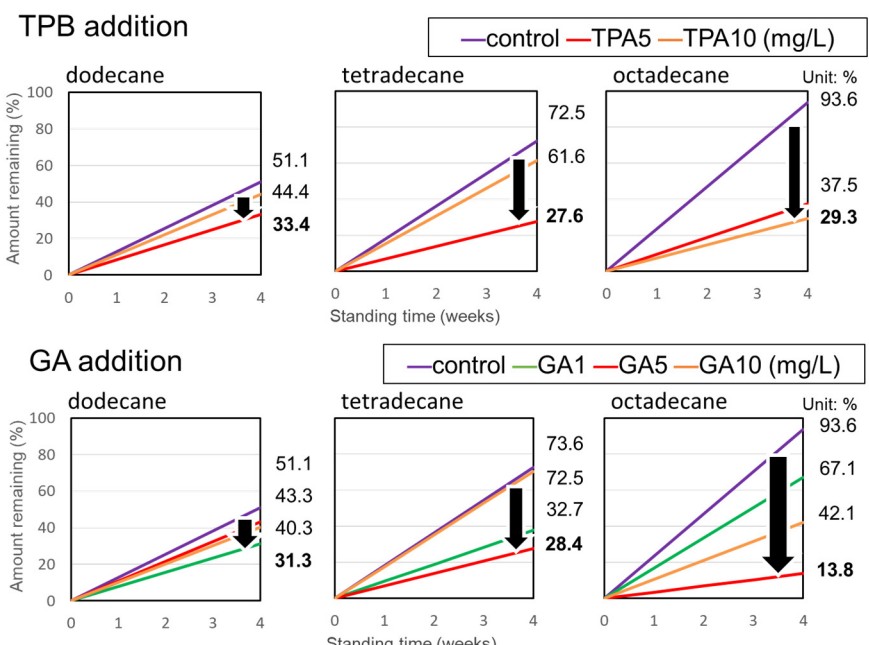

**Figure 7.** Changes in the amount of linalooloxide.

### 3.4. Antimicrobial Effectiveness Test

Next, the antimicrobial effects of the water-soluble extracts of immature persimmons are shown.

We tested the five types of compounds extracted from immature persimmons on five types of bacterial strains that are listed in Table 5. For comparison, we used persimmon tannins, tannic acid, gallic acid, and catechin mixture.

**Table 5.** Used bacterial strains and samples.

| Bacterial Strain |
| --- |
| *Bacillus subtilis* NBRC3134 |
| *Escherichia coli* NBRC3301 |
| *Klebsiella pneumoniae* NBRC13277 |
| *Pseudomonas aeruginosa* NBRC13275 |
| *Staphylococcus aureus* NBRC13276 |
| **Sample** |
| FFA (Fuyu persimmon flesh crude extract) |
| FFW (Fuyu persimmon flesh water extract) |
| FFB (Fuyu persimmon flesh butanol extract) |
| TFB (Tone Wase flesh butanol extract) |
| TPB (Tone Wase peel butanol extract) |
| Persimmon tannin |
| Tannic acid |
| Gallic acid |
| Catechin mixture |

Table 6 shows the results of the antimicrobial effectiveness tests. Minimum inhibitory concentrations were obtained in a 96-well microplate. Unfortunately, the highest concentration of the reagent used in this experiment (500 mg/L) was unable to inhibit the growth of the bacteria with the components extracted from immature persimmons.

**Table 6.** Antimicrobial effectiveness test.

| Sample | *B. subtilis* | *E. coli* | *K. pneumoniae* | *P. aeruginosa* | *S. aureus* |
| --- | --- | --- | --- | --- | --- |
| FFA | >500 | >500 | >500 | >500 | >500 |
| FFW | >500 | >500 | >500 | >500 | >500 |
| FFB | >500 | >500 | >500 | >500 | >500 |
| TFB | >500 | >500 | >500 | >500 | >500 |
| TPB | >500 | >500 | >500 | >500 | >500 |
| Persimmon tannin | 250 | >500 | >500 | >500 | 125 |
| Tannic acid | 125 | >500 | >500 | >500 | 31 |
| Gallic acid | >500 | >500 | >500 | >500 | 125 |
| Catechin mixture | >500 | 500 | 500 | 250 | 62 |

MIC (Minimum inhibitory concentration) mg/L.

## 4. Discussion

All the isolated compounds **1** to **12** showed higher activity than the positive control aminoguanidine in inhibiting the formation of AGEs. In particular, compounds **6** to **10** showed very high activity. These were catechins and procyanidins, which are polymers of catechin.

Dicarbonyl compounds, such as glyoxal and methylglyoxal, are intermediates in the AGE formation route. It has been reported that catechins mainly trap dicarbonyl compounds through either the C6 or C8 position of the A-ring [13]. Moreover, free radicals are also closely related to the formation of AGEs because their formation process generates a large number of free radicals [14]. Free radicals may mediate the conversion of Amadori compounds to AGEs [15]. It has been reported that catechins display potent scavenging activity towards free radicals when compared to the positive control aminoguanidine, removing free radicals mainly through the phenolic hydroxyl group of the B-ring [16]. These findings suggest that catechins are involved in the inhibition of AGE formation

at each stage of the route, ranging from the Amadori compounds (intermediates) to the formation of AGEs.

Furthermore, the antioxidant activity of procyanidin varies depending on the degree of polymerization, the binding site, and the tertiary structure of the polymer, and the antioxidant capacity increases as the degree of polymerization rises [17]. Procyanidin gives a hydrogen atom to the phenolic hydroxyl group of the B-ring. This blocks chain oxidative reactions by acting as a free radical scavenger that quenches free radicals [18]. Procyanidin is thought to be involved in the inhibition of AGE formation at each step because the AGE formation route involves stepwise oxidative reactions.

Here, the structure-activity relationship of the isolated compounds reveals that compound **7**, a dimer of compound **6** (catechin), shows higher activity than compound **6**. There is a difference between compounds **8** and **9** in that the former has protocatechuic acid attached to procyanidin B, and the latter has gallic acid attached to procyanidin B. However, compound **9**, which has more hydroxyl groups, showed higher activity. In addition, compound **10** was a trimer with the highest degree of polymerization and the highest activity among these compounds.

In the comparison of extracts, when the extracts of immature persimmons were compared with those of ripe persimmons, the activity in Tone Wase persimmons that had not undergone astringency removal was higher than that in immature persimmons. However, this activity was greatly reduced by astringency removal. Regarding Fuyu persimmons, activity in ripe persimmons was lower than that in immature persimmons. As a result, activity in immature persimmons can be considered to be higher than that of the persimmons that we usually eat. This is thought to be due to the effect of tannins. First, when the Fuyu (sweet persimmon) ripens, acetaldehyde generated inside the fruit forms polymers by cross-linking the soluble tannins, making the tannins insoluble. The Tone Wase (astringent persimmon) has a high soluble tannin content of 1–2% when ripe. The anaerobic treatment of persimmon with carbon dioxide or ethanol increases the acetaldehyde content, rendering the tannins insoluble.

Based on the above and the other results of this study, high activity was observed in immature persimmons containing soluble tannins and in ripe Tone Wase persimmons that had not undergone astringency removal, but activity was not observed in ripe Tone Wase persimmons that had undergone astringency removal and contained insoluble tannins or in ripe Fuyu persimmons. Therefore, it is presumed that soluble tannins have an inhibitory effect on AGE formation, while insoluble tannins may not have such activity.

In the beverage and flavoring experiments, the extracts of immature persimmons and gallic acid had no inhibitory effect on deterioration. For TPA, deterioration was inhibited at lower sample concentrations except for dodecane, and for GA, the concentration that showed the greatest inhibition was 5 mg/L. It is believed that low concentrations did not provide sufficient antioxidative potency to protect the citral and failed to inhibit its deterioration. The following two points can be considered as possible reasons for the failure to sufficiently inhibit deterioration when concentrations became too high. First, it is thought to be due to the acidic nature of the sample, which led to its deterioration. Other causes include the fact that gallic acid is also an acid, and the extracts of immature persimmons that contain it are also thought to be acidic. The deterioration may have been accelerated due to the more acidic nature of the sample added as its concentration increased. Second, pro-oxidants may be produced due to the generation of other radicals associated with the elimination of oxidizing radicals. In a previous study, the inhibition of the deterioration of citral was investigated using ubiquinol as an antioxidant. The results showed that neither too low nor too high a concentration sufficiently inhibited deterioration, indicating the existence of an optimal concentration [19]. Although it is difficult to elucidate the detailed mechanism, similar phenomena have been observed for other antioxidants, such as carotenoids [20].

In addition, there were no deteriorated substances such as p-methyl acetophenone (Figure 2), which exhibited a petroleum scent, and the deteriorated substances formed

were myrcenol and linalool oxide (Figure 4). As neither of the two deteriorated substances was cyclized, it is thought that the deterioration of p-menthadien-8-ol was inhibited at the preliminary stage of its formation. In the presence of the emulsifier SY-Glyster ML-750, citral was converted into alcohols by the Cannizzaro reaction and proceeded to become linalool, which was ultimately converted into linalooloxide and myrcenol. Furthermore, GA 5 mg/L in octadecane was the most effective inhibitor of the decrease and deterioration of citral in this study. According to research collaborator Prof. Yoshiyuki Watanabe, it was reported that when the emulsions were stored at 37 °C, the size of oil droplets increased and the amount of limonene in the oil phase of the emulsion decreased with time, and the limonene disappearance rate generally decreased as the carbon number of the oil phase alkane increased [11]. Octadecane had the highest carbon number among the alkanes used in this study, suggesting that octadecane may have been the condition under which citral was the least likely to be lost from the emulsion. Consequently, GA 5 mg/L in octadecane could be considered as the optimal stability condition for the storage stability of emulsions and oxidative stability of citral emulsions. Based on these findings, we made a microencapsulation process, and plan to advance to the clinical trial study in future.

## 5. Conclusions

Immature persimmons are unripe fruits that are cut off during the persimmon cultivation process and immediately discarded, amounting to an annual fruit loss of approximately 100 to 400 kg per 1000 m$^2$. The purpose of this study was to make effective use of unused resources, namely, immature persimmons, and attempt to use them as food additives. In this study, we studied the Tone Wase (fully astringent persimmon) and Fuyu (fully sweet persimmon) cultivars. As a result, we performed a component analysis of the immature persimmons, isolating 12 compounds, of which 2 were newly identified. Differences in the components and their contents were found between cultivars and between the peel and flesh.

To effectively use immature persimmons as food for the elderly, we searched for active substances that inhibit AGE formation and found that extracts of immature persimmons and isolated compounds showed high activity. In particular, high activity was observed for catechin and its polymeric form, procyanidin. Regarding the inhibition of aroma deterioration, 5 mg/L of gallic acid in octadecane was found to be the optimal condition for the inhibition of citral deterioration. As for antimicrobial activity, we found that extracts at a concentration of 500 mg/L had no antimicrobial effect. Based on these findings, we made a microencapsulation process, and plan to advance to the clinical trial study in future.

These findings confirmed the effectiveness of immature persimmons, which are an unused resource, and reveal their potential as a food for the elderly and as a food additive in other food products, which we hope will lead to new industrial innovations.

**Author Contributions:** A.S. and Y.W. conceived and designed the research. N.O. carried out all experiments. S.Y. carried out the experiment of antimicrobial effectiveness test. K.K. (Kunihiro Kusudo) carried out the experiment of the deterioration inhibitory effect on citrus fruit flavor. K.K. (Kenichi Kanno) performed the simulations. All authors have read and agreed to the published version of the manuscript.

**Funding:** A part of this research was funded by the 21st Century Joint Research Enhancement Grant from Kindai University, grant number KD201705 and KD2003.

**Institutional Review Board Statement:** Not applicable.

**Informed Consent Statement:** Not applicable.

**Data Availability Statement:** The data presented in this study are available on request from the corresponding author. The data are not publicly available due to privacy.

**Acknowledgments:** We thank Mitsuaki Yamashita for the stimulating discussion.

**Conflicts of Interest:** The authors declare no conflict of interest.

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
