# Peer review of "Functional Ingredients and Food Preservative in Immature Persimmon “Tekka-Kaki”"

_processes, doi:10.3390/pr9111989_

Round 1
Reviewer 1 Report
- The abstract needs to be completely rewritten as the language was very bad. it's very hard to understand, unlike the text.
- The background mentioned about microencapsulation of the fruit, but completely missing in the discussion and conclusion.
- The peel is not mentioned at all in the background, surprised to suddenly read about it in the method section.
- The method needs to be supplied with more details. For example 1. the flow rate for the gel filtration, 2. the equipment for the column chromatography.
- Why 2 analytical instruments are needed and which analysis for which instrument?
- The conclusion need to be reworked. Didn't really answer the research gap mentioned in the background.
Author Response
Thank you very much for valuable suggestions and comments to our manuscript. According to the suggestions of the reviewer, we prepared a revised manuscript. Followings are major points of revision and our comment.
We agree with the suggestion of reviewer. Therefore, We changed everything according to the suggestion of reviewer.
1. The abstract needs to be completely rewritten as the language was very bad. it's very hard to understand, unlike the text.
--> According to suggestion, we revised the abstract whole sentence.
2. The background mentioned about microencapsulation of the fruit, but completely missing in the discussion and conclusion.
--> Lines 594 and 612 We add "Based on these findings, we make microencapsulation process and are advance to the clinical trial study in future.".
3. The peel is not mentioned at all in the background, surprised to suddenly read about it in the method section.
--> Line 60 We add "the peels of ".
4. The method needs to be supplied with more details. For example 1. the flow rate for the gel filtration, 2. the equipment for the column chromatography.
--> Line 185-220 We revised the whole sentence of 2.2, 2.3 and 2.4.
5. Why 2 analytical instruments are needed and which analysis for which instrument?
--> Line 222 According to suggestion of Reviewer 2, we revised the section title to Mass spectrometry and NMR analyses.
6. The conclusion need to be reworked. Didn't really answer the research gap mentioned in the background.
--> Line 612 We add "Based on these findings, we make microencapsulation process and are advance to the clinical trial study in future.".
Reviewer 2 Report
The manuscript titled 'Functional ingredients and food preservative in immature persimmon "Tekka-kaki" ' is carried out with the aim of isolating compounds from the persimmon fruit which demonstrates anti-aging effects and some anti-bacterial potential. They isolated several compounds from the fruits and formulated an emulsion. Written nicely but needs some correction.
Line 41: mention the years range for the "Nara period"; Mention some of the roles of the persimmon fruit before talking about the astringency of the fruit, for eg. some characteristics.
Lines 89-94: Add references to support these lines.
Line 107: italicize "in vitro"
Line 156: 'It is divided'
Line 184: change section title to Mass spectrometry and NMR analyses.
Line 300: How was the emulsion prepared, which apparatus was used?
Line 319: mention the name of the bacteria.
Line 345: The NMR and MS spectra should be presented.
Supplementary materials missing from the text
Author Response
Thank you very much for valuable suggestions and comments to our manuscript. According to the suggestions of the reviewer, we prepared a revised manuscript. Followings are major points of revision and our comment.
We agree with the suggestion of reviewer. Therefore, We changed everything according to the suggestion of reviewer.
Line 41: mention the years range for the "Nara period"; Mention some of the roles of the persimmon fruit before talking about the astringency of the fruit, for eg. some characteristics.
--> Line 59 According to suggestion, we add ", 710-794 years".
Lines 89-94: Add references to support these lines.
--> Line 111 According to suggestion, we add reference [20].
Line 107: italicize "in vitro"
--> Line 126 According to suggestion, we italicize "in vitro"
Line 156: 'It is divided'
--> Line 176 According to suggestion, we add “is”.
Line 184: change section title to Mass spectrometry and NMR analyses.
--> Line 222 According to suggestion, we revised “Mass spectrometry and NMR analyses”.
Line 300: How was the emulsion prepared, which apparatus was used?
--> Line 341 According to suggestion, we add reference [19].
Line 319: mention the name of the bacteria.
--> Line 500 We showed the name of the bacteria in table 5.
Line 345: The NMR and MS spectra should be presented.
--> Line 235-319 We showed the NMR and MS data in section 2.6. Isolated compounds.
Reviewer 3 Report
Interesting paper. A native English speaker should correct the text, especially the abstract.
Author Response
Thank you very much for valuable suggestions and comments to our manuscript. According to the suggestions of the reviewer, we prepared a revised manuscript. Followings are major points of revision and our comment.
We agree with the suggestion of reviewer. Therefore, We changed everything according to the suggestion of reviewer.
A native English speaker should correct the text, especially the abstract.
--> According to suggestion, we revised the abstract whole sentence.